

# Multiscale analysis of nitrogen adsorption and desorption isotherms in soils with contrasting pedogenesis and texture

Jorge Paz-Ferreiro[1], Mara de A. Marinho[2], Cleide A. de Abreu[3], Eva Vidal-Vázquez[4]

[1]Royal Melbourne Institute of Technology University, School of Civil, Environmental and Chemical Engineering, Melbourne, Australia.

[2]Faculdade de Engenharia Agricola (FEAGRI), Universidade Estadual de Campinas (UNICAMP), Av. Candido Rondon, 501, Campinas, 13083-875, SP, Brazil.

[3]Instituto Agronômico de Campinas (IAC), Av. Barão de Itapura, 1481, Campinas, 13020-902, SP, Brazil.

[4]Facultad de Ciencias, Universidade da Coruña, Campus de Elviña, sn. Coruña, Spain.

*Correspondence to*: jpaz@udc.es

**Abstract.** The specific surface area (SSA) of a soil is commonly estimated from adsorption isotherms determined in a limited range of relative pressures ($p/p_0$), admitting a non fractal model. Nitrogen adsorption (NAI) and desorption (NDI) isotherms determined over the full range of $p/p_0$ have been described using the multifractal approach. This study aimed to assess effects of soil texture on the multifractality of NAIs and NDIs, and to analyze the association between multifractal parameters and soil properties. Six soil profiles were taken to get two groups of samples with contrasting pedogenetic origin, texture (medium or clayey), susceptibility to water erosion and quality for agricultural uses. These two soil groups also were significant differences in SSA and cation exchange capacity (CEC), but not in organic matter content (OMC). Consistent with previous studies, the scaling properties of both NAIs and NDIs from all the soil horizons studied could be fitted reasonably well with multifractal models. Values of parameters $D_{-5}$, $D_1$, $D_2$ and $D_5$, extracted from the generalized dimension function, $D_q$, were higher for clayey soils during adsorption, but during desorption all of them were higher for medium textured soils. Therefore, the measure was more evenly distributed for clayey soils during adsorption and for medium textured soils during desorption. Width of $D_q$ function given by parameter ($D_{-5}$-$D_5$) was significantly higher clayey soils for NAIs, but not significant differences were detected for NDIs; subsequently scaling heterogeneity of NAIs was higher for clayey than for medium textured soil. Differences in multifractal behaviour of NAIs and NDIs were consistent with a wider hysteresis loop of the medium texture soils compared to that of the clayey soils. Linear correlations were found between parameters $D_{-5}$ and ($D_{-5}$ - $D_5$) and clay content or SSA, which were positive and negative for NAIs and NDIs, respectively. Agronomical and environmental characterization of these soil groups with contrasting properties be enhanced by evaluating SSA and by inspecting NAIs and NDIs for multifractality.

Key words: nitrogen isotherms, soil specific surface, multifractals, texture, soil composition soil use.

## 1. Introduction

The quality of a soil, defined as its ability to perform a given function or its suitability for chosen uses in agroecosystems, depends both on inherent or dynamic soil properties (Doran and Parker, 1994; Carter et al., 1997, Lal, 1998). Inherent soil properties such as particle size distribution, particle density, or soil mineralogy rely upon soil-forming factors, whereas dynamic soil properties, such as aggregate stability, water and nutrient status, bulk density change in response to soil use and


management (Carter et al., 1997), bur also may be affected by inherent soil properties. Several properties such as organic matter content, SSA or bulk density may be considered as inherent properties for deep horizons, but have been shown to be dynamic, or use dependent, near the soil surface.

The soil mineral fraction is most frequently characterized by particle size analysis, because this is an inherent soil property, which greatly influences the physical and chemical processes, which affect soil functions. Also, several macro-scale physical and chemical soil properties and processes are closely related to grain-scale properties, such as SSA, porosity, pore size distribution, pore geometry and energy distribution (Petersen et al., 1996, Hajnos et al., 2000), which are use dependent at least at the top soil horizons. In particular, specific surface area (SSA) has been commonly considered as an important soil property, which is strongly related to soil texture, clay type, reactivity of soil colloids and retention or release of chemicals (Hepper et al., 2006; Feller et al., 1992; Vidal-Vázquez and Paz-Ferreiro, 2012; Lado et al., 2013).

Determinations of either adsorption isotherms or both, adsorption-desorption isotherms, at constant temperature, are usually performed to estimate soil surface properties, including SSA and soil porosity. The adsorbate most frequently used to obtain these isotherms is gas Nitrogen (Rouquerol et al., 1999). The most common surface property evaluated from Nitrogen sorption isotherms is SSA. This is because soil SSA has been proven to be a useful soil property, which has been correlated with important soil texture, soil structure, soil mineralogical composition, exchangeable cations, water retention, etc. (Petersen et al., 1996; Hajnos et al., 2000; Jozefaciuck et al., 2006; Bartoli et al., 2007; Paz-Ferreiro et al., 2009, 2013; Lado et al., 2009). To estimate SSA, classical, non-fractal models are employed, from which the most conventional is the Brunnauer-Emmett-Teller (BET) model (Brunauer et al., 1938). NAIs and NDIs determinations can be easily performed over the entire range of relative pressure, $0 < p/p_o < 1$, and provide much more information that necessary for SSA estimations, which only require the data contained in a limited range of $p/p_o$.

Fractal-based models have been in the past used to describe soil NAIs (Pachepsky et al., 1995; Hajnos et al., 2000; Jozefaciuk et al., 2006). Also, the scaling properties of NAIs from soils (Paz-Ferreiro et al., 2009, 2010, Vidal-Vazquez and Paz-Ferreiro, 2012) and artifitial organoclays (Lado et al., 2013) have been reasonably well described by multifractal models. More recently, Paz-Ferreiro et al., 2013 performed multiscale analysis of both NAIs and NDIs. Indeed, comparison of results from the classical BET model and those from multifractal approaches is not straightforward. First, the BET model, estimates the total surface area from adsorption isotherms in a limited range of relative pressure, (i.e., $0.05 < p/p_0 < 0.35$), while fractal and multifractal approaches use the information contained in the entire adsorption or desorption curve. Second, the BET method assumes that the soil pore-solid interface is not a fractal (Paz-Ferreiro et al., 2009). Nevertheless, it has been claimed that SSA and scaling analysis of $N_2$ isotherms yield complementary information that may be useful for a better understanding of the geometry of soil surfaces and porous systems (Paz-Ferreiro et al., 2013).

Until now, multifractal analysis of NAIs (Paz-Ferreiro and Vidal Vázquez, 2012) and both, NAIs and NDIs (Paz-Ferreiro et al, 2013) has been carried out in Brazilian soils, collected in Minas Gerais and Santa Catarina states, respectively. Soil samples in this later work were mostly clayey textured, but had a wide range of soil organic carbon (SOC) content; therefore its main focus was on the interaction between SOC and the scaling property of NAIs and NDIs. For the present study, contrasting, medium textured or clayey textured, soil profiles were sampled in neighbouring sites of São Paulo State, Brazil. The two different textures are the result of the interaction between pedogenesis and morphogenesis across the landscape. In particular, parent materials and topography have been shown to be the main soil-forming factors that explain soil distribution and soil properties at the local scale (Oliveira et al., 1979; IPT, 1981, 1997). Most of the State of São Paulo belongs to the Brazilian Central Highlands or Central Plateau, which consists of a rugged tableland with smooth undulated relief varying in altitude from 400 to 700 meters a.s.l. Medium textured soils prevail on unstable, dissected areas, while heavily textured soils



prevail on stable, tableland landscape. Medium textured soils are developed from parent material, rich in silica and poor in basic cations, while clayey soils are formed over strongly weathered, allochthonous materials, with a higher base status. Subsequently, the selected medium and clayey textured soil groups also are characterized by dissimilar physical chemical and biological properties, and therefore very distinct susceptibility to erosion and quality for agricultural uses (Weill and

Sparovek, 2008).

Understanding inherent and use dependent soil properties of these soils already has been demonstrated to be useful to prevent misuse and land degradation, contributing to agriculture sustainability and subsequently to promove the protection of environmental quality. Although the studied soils have been extensively investigated in terms of soil genesis, soil properties and soil uses and management, little is known about SSA and other soil surface properties. Here, we hypothesized that

analysis of the information

contained in NAIs of NDIs of these two contrasting soil groups may provide further insight for its agronomical and environmental characterization. Therefore, the two main objective of this work were: i) to examine and to compare the scaling property of NAIs and NDIs in soils with contrasting texture and ii) to analyze the association between multifractal parameters and soil properties, focussing on soil texture and clay content. Additionally, we evaluate SSA by the classical

BET model and assessed its relationships with general soil properties and we discussed the potential usefulness of the data sets obtained to further characterize these soils.

## 2.  Materials and Methods

### 2.1.  Site characteristics and soil sampling

The study was conducted at the region of Campinas, São Paulo State, Brazil. Site altitudes ranged from 574 to 640 m above

see level. According to Köppen the local climate is a transition between two mesotermic types, those with dry winter (Cwa) and hot summer (Cfa). Mean annual temperature in Campinas is $22.4^{\circ}$C and a mean yearly precipitation 1382 mm.

Six soil profiles were selected and sampled; three of them were developed over sedimentary rocks with high silica contents (sand- and siltstones), and the other three were over strongly weathered and reworked material with a higher proportion of basic mineral. Table 1 lists depth of the 32 horizon collected from the 6 soil profiles, main site characteristics (location,

parent material) and classification, following the Brazilian System of Soil Classification, BSSA, described by EMBRAPA (2013), Soil Survey Staff, SSS (2014) and World Reference Base, WRB, (2006). Soil profiles nº 1 to 5 were sampled in municipalities neighboring to Campinas, while profile nº 6 was sampled at the experimental farm of the College of Agricultural Engineering, State University of Campinas (FEAGRI-UNICAMP). Location of profiles 1 to 5 can be seen as Supplemental Digital Content, Figure S1.

Profiles 1, 2 and 3, (in short P1, P2 and P3), sampled at the Monte Mor municipality, were over fine sandstones and siltstones from the Tubarão formation (Upper Carboniferous), and they belong to a toposequence developed along a hillside on undulate to strong undulate relief. P1 was located on the lower steep hillside and it was classified as an Udorthent, while P2 and P3 were sampled at the middle and upper hillside, respectively and classified as Hapludults (SSS, 2014).These soils were devoted to extensive pasture. Typically they are characterized by a high erodibility index, and subsequently a high

susceptibility to water erosion, which is enhanced by the rolling topography and by the presence of a lithic contact (P1), or a textural gradient (P2) that may be abrupt (P3).



Profiles 4 and 5 (in short P4 and P5), collected at Sumaré Municipality, were developed on strongly weathered and reworked deposits consisting of a mixture of loamy-clayey sediments and diabase materials, respectively; also profile, P6, sampled at the FEAGRI campus, Campinas, was over deeply, weathered, reworked diabase on a smooth slope position. Profile P4, classified as an Hapludox was located on the flatter plateau at the top of the hillside, while P5, classified as a Rhodudult, (equivalent to Nitisol in the WRB) was on slope position at the middle hillside; finally P6, was on a smoother slope and also it was classified as an Hapludox. P4 and P5 were cropped to sugar cane, while P6 was used for crops in rotation. Soils over deep weathered materials (P4. P5 and P6) have been described as well developed soils, with stable and good functional structure and therefore low erodibility and susceptibility to water erosion. (Weill and Sparovek, 2008).

### 2.2. Analysis of general soil physical and chemical properties

Soil samples were ground to pass through 2 mm sieve. For each soil horizon collected, clay, silt and sand contents were determined by the sieve-pipette method (EMBRAPA, 1997; Dorado et al., 2013). Determinations of pH, organic carbon content, exchangeable bases (Ca, Mg and K) and exchangeable acidity (H + Al) were conducted as described in van Raij et al. (2001). Exchangeable bases and exchangeable acidity ere also used to calculate cation exchange capacity (CEC), sum of bases (SB) and percent base saturation (V %) for each sample.

### 2.3  $N_2$ isotherms and soil specific surface area

Determinations of nitrogen adsorption and desorption isotherms were obtained in A Soptomatic 1990 equipment manufactured by Thermo Finnigan (Milano, Italy). Two replicate measurements per horizon were performed in small aggregates. The inert gas used ($N_2$) was 99.998% pure and determinations were performed at the liquid state (77 K temperature). Sample preparation and determination of NAI and NDI have been previously described, and details can be found in Paz Ferreiro et al., 2009, 2013. Adsorption isotherms were acquired in a scale of relative pressures, $p/p_0$, ranging from 0.001 to about 0.997 (as in Paz-Ferreiro et al., 2009; Vidal-Vazquez and Paz-Ferreiro, 2012). The scale used for desorption isotherms was from the highest relative pressure of about 0.997 to lowest $p/p_0$ values near 0.01 (as in Paz-Ferreiro et al., 2013).

The soil SSA was estimated from the adsorption branch of the isotherms in the range low relative pressure values ($0 < p/p_0 < 0.35$), using the BET model (Carter et al., 1986). This model easily estimates the total surface area of a soil sample, since the area covered by a single molecule adsorbed on the soil surface is known; implicitly, it assumes that surface and pore geometry is Euclidean (Rouquerol et al., 1999; Bartoli et al., 2007, Paz-Ferreiro et al., 2013). Figure 1 shows examples of absorption-desorption isotherms from selected horizons over sandstone and weathered materials.

### 2.4  Multifractal analysis

The method of moments (Halsey et al., 1986) and the direct method (Chhabra and Jensen, 1989) were employed here to perform multifractal analysis of NAIs and NDIs. This procedure has been frequently used for multifractal evaluation of various soil properties, including pore size distributions (Posadas et al., 2003; Tarquis et al., 2006), Particle size distributions (Miranda et al., 2006) surface roughness (Vidal Vázquez et al., 2008) etc. Also, more recently it has been employed for assessing multifractal property of either the adsorption branch (Paz-Ferreiro et al., 2009, 2010; Vidal-Vázquez and Paz-Ferreiro, 2012; Lado et al., 2013) or both, the adsorption and desorption branches (Paz-Ferreiro et al., 2013) of nitrogen isotherms. Therefore multifractal concepts and method of analysis will be here only briefly described.





Cumulative adsorption and desorption data sets are taken as raw data, from which differential change of $N_2$ volume, $\Delta n$, for each $p/p_0$ interval can be computed. Therefore, the distributions of $N_2$ was taken as the measure, $\mu_i$, and the relative pressure, $p/p_0$, itself, as the support. To evaluate the scaling behavior of the measure, $\mu_i$, the support $p/p_0$ was next divided into successive segments with a unit length, $\delta$, following dyadic downscaling (Paz Ferreiro et al.; 2009, 2013)

Thereafter, the data sets consisting of distributions of $N_2$ during absorption or desorption are normalized, meaning that a new variable, the probability mass function, $p_i(\delta)$ or $\mu_i(\delta)$, is defined as:

$$p_{ii}(\delta) = \mu_i(\delta) = \frac{N_i(\delta)}{N_t}, \tag{1}$$

where $N_i(\delta)$ is the value of the measure in the $i^{th}$ segment of scale $\delta$ in a $p/p_0$ interval and $N_t$ represent the total mass in the whole scale of applied relative pressure .

Multifractal analysis of the probability mass function yields the following functions: mass exponent function, $\tau_q$, generalized dimension, $D_q$. and singularity spectrum, $f(\alpha)$ versus $\alpha$. First, a partition function, $\chi(q,\delta)$, was estimated from the $p_i(\delta)$ values as defined as: $\chi(q,\delta) = \sum_{i=1}^{n(\delta)} \mu_i^q(\delta)$, where $n(\delta)$ is the number of intervals covering the $p/p_0$ scale and $q$ is the order of the statistical moment. The partition function scales with the box size, $\delta$, as:

$$\chi(q,\delta) \propto \delta^{-\tau(q)} \tag{2}$$

where $\tau(q)$ is the mass exponent or scaling function of order $q$.

The scaling function $\tau_q$ is also related to the generalized dimension $D_q$, Therefore, multifractal sets can also be characterized by their spectrum of generalized dimensions using the following relationships:

$$D_q = \lim_{\delta \to 0} \frac{1}{q-1} \frac{\log[\chi(q,\delta)]}{\log \delta} = \frac{\tau_q}{(1-q)}, \; q \neq 1 \tag{3a}$$

$$D_1 = \lim_{\delta \to 0} \frac{\sum_{i=1}^{n(\delta)} \mu_i(\delta) \log[\mu_i(\delta)]}{\log \delta}, \; q = 1 \tag{3b}$$

The generalized dimensions, $D_q$ for $q = 0$, $q = 1$ and $q = 2$, are known as the capacity, the information (Shannon entropy) and correlation dimensions, respectively. The spectra of generalized dimensions for different $q$ have specific features for multifractals (i.e. $D_0 > D_1 > D_2$), while for monofractals $D_q$ is a constant.

The singularity spectrum, $f(\alpha)$, and the coarse Hölder exponent, also known as local scaling index, $\alpha_q$, can be estimated from the mass exponent function, $\tau_q$ through a Legendre transformation. However, this procedure is not straightforward, and most

frequently $f(\alpha)$ and $\alpha$ have been obtained by the direct method.

The direct method (Chhabra and Jensen, 1989) employs the scaling properties of another normalized variable, and is based on the contributions of individual segments to the partition function, $\mu_i(q,\delta)$, that is defined as:

$$\mu_i(q,\delta) = \mu_i^q(\delta) / \sum_1^{n(\delta)} \mu_i^q(\delta). \tag{4}$$

Now, using a set of real numbers, $-\infty < q < \infty$, the functions $f(\alpha)_q$ and $\alpha_q$ can be computed as follows::



$$f(\alpha(q)) \propto \frac{\sum_{i=1}^{N(\delta)} \mu_i(q,\delta) \log[\mu_i(q,\delta)]}{\log(\delta)}$$

(5a)

$$\alpha(q) \propto \frac{\sum_{i=1}^{N(\delta)} \mu_i(q,\delta) \log[\mu_i(\delta)]}{\log(\delta)}$$

(5b)

As before stated, the scale of experimental NAI and NDI curves was in the range of relative pressures: $0.001 < p/p_0 < 0.997$ and $0.01 < p/p_0 < 0.997$. The first points of the scale were accepted as similar for adsorption and desorption phases. Using

this rule, the number of experimental data points of $N_2$ volume versus relative pressure ($p/p_0$) was between 41 and 52.

Linearity of these log-log plots of the normalized measures $\chi(q,\delta)$ versus measurement scales, $\delta$, was found for successive partitions from $1 < k < 4$, as in a previous study (Paz-Ferreiro et al., 2009). For $k < 1$, however, the double logarithm plots departed from linearity. Generalized dimension spectra, $D_q$, were calculated with Eq. (6) in the moment range $-5 \leq q \leq 5$ at 0.5 lag increments. Values $\alpha$ and $f(\alpha)$ of the singularity spectrum were calculated using Eq. (5). Points ($\alpha$, $f(\alpha)$) were

accepted in the singularity spectrum only if the logarithm of the normalized measures varied linearly with the logarithm of the measurement scale, which means regressions with coefficients of determination, $r^2 \geq 90$. Subsequently, Several parameters were obtained form the generalized dimension spectra for successive $q$ moments (i.e., $D_5$, $D_0$, $D_1$, $D_2$, $D_{-5}$) and the singularity spectra (i.e $\alpha_0$ or Hölder exponent of order zero).

### 2.5  Statistical analysis

One way ANOVA was carried out to compare general properties and multifractal parameters among soil groups and between NAIs and NDIs. Differences between mean values of these variables at the $P<0.05$ level were tested using the Fisher Least Significant Differences (LSD) procedure and the Tukey test.

Product-moment correlations were performed between soil physico-chemical properties, and multifractal parameters. Principal component analysis (PCA) was performed for data set consisting of soil physico-chemical properties and several

multifractal parameters cropped from NAIs and NDIs. All the raw data were standardized for mean 0 and variance 1 and PCA was performed in the resulting data matrix. The three first principal components (PC1, PC2 and PC3) were selected for the ordination of cases. Statistical analyses were performed using SAS scientific software, version 8.0 (SAS, 1999).

## 3  Results and discussion

### 3.1  General soil physico-chemical and surface properties

General soil physical and chemical properties of the studied soils are listed in Table S1 of the Supplementary Digital Content. Profiles 1 to 3, over sand- and siltstones, were  loamy and sandy loam textured, while profiles 4 to 6, over strongly weathered, reworked materials were clayey textured, except for the top horizon of profile 4, which was sandy clay. Clay content in the former group of soil profiles was lower than 225 mg kg$^{-1}$, whereas it was higher than 384.5 mg kg$^{-1}$ for the latter group. For simplicity, these two soil groups with contrasting clay contents will be next referred to as medium textured

and clayey textured soils.

Organic matter contents for medium textured and clayey soils ranged 16-37 g kg$^{-1}$ and 16-31 g kg$^{-1}$, respectively. Medium textured soils over sedimentary rocks wit high silica content had pH values from 4,1 to 4.9, while the counterpart over more



basic materials ranged 4.1-5.6. The two groups of soils studied were characterized by low CEC values, (< 13 $Cmol_+$ $kg^{-1}$). Notwithstanding, CEC was significantly higher for clayey soils than for medium textured soils (P < 0.05), and this trend was also observed for Al + H. However, exchangeable K, Mg, Mg as well as sum of exchangeable bases, SB and percent base saturation, V, showed similar values in these two soil groups; specifically, SB values were rather low (< 4 $Cmol_+$ $kg^{-1}$) in the

two soil groups.

Differences in nitrogen isotherms of the two soil groups are noteworthy, as shown in Figure 1. Here, the cumulative volume of $N_2$ adsorbed was about 15 times higher for the selected clayey horizon, compared to the loamy counterpart. The hysteresis loop, however, clearly was wider in the loamy horizon than in the clayey horizon.

Values of SSA were in the range from 2.86 to 47.26 $m^2g^{-1}$. Conform to clay contents, SSA was below 15.09 $m^2$ $g^{-1}$ for

medium textured soils, and above 26.21 $m^2$ $g^{-1}$ for clayey soils (Figure 2). Overall, clay content and SSA showed a very strong correlation (r > 0.99, P<0.01, see also Table 1). The regression equation between SSA and clay content for our studied soils was: SSA = 0.75 clay -1.26, quite similar to that proposed by Feller et al. (1992) for tropical soils. However, SSA values of soils from São Paulo in this study are lower than those reported fro soils from Minas Gerais (Vidal Vázquez and Paz-Ferreiro, 2010) and Santa Catarina (Paz-Fereiro et al., 2013).

In addition, no significant relationship was found between these soil SSA and properties of the soil exchange complex (CEC, SB or exchangeable cations). The association between SSA and CEC has been proved for soils from temperate climates (Petersen et al., 1996; Hepper et al., 2006; Paz-Ferreiro et al., 2009). However, tropical soils frequently have been identified by a clay fraction containing not only clay minerals, but also rich in oxides hydroxides of iron and aluminium (Feller et al., 1992). These secondary constituents present in the clay fraction may contribute to SSA, but are not able to develop

significant CEC.

ANOVA analysis showed significant differences (P<0.05) between medium textured versus clayey soils, (i.e. P1 to P3 versus P4 to P6 ), for mean values of texture fractions (sand, silt and clay), SSA, CEC and H+Al, while mean values of pH, organic matter content, SB and V were statistically similar (data no shown).

### 3.2   Multifractality of adsorption and desorption isotherms

Because partition functions have been estimated in the range of linear behaviour, involving segment sizes limited to 1< k <4, the range of log (δ) employed in this study was between 0.30 and 1.40. Partition functions in our work are similar to those shown in Paz-Ferreiro et al. (2009) and Vidal Vázquez and Paz Ferrero, (2012).

Table 3 list various multifractal parameters ($D_{-5}$, $D_1$, $D_2$, $D_5$) extracted from the generalized dimension function, $D_q$ versus q, as well as parameter α from the singularity spectrum, f(α) versus α, of adsorption isotherms of the studied soils. Table 4 lists

the respective parameters for desorption isotherms. Examples of $D_q$ versus q functions are shown in Figures 3 and 4.

The generalized dimension spectrum, $D_q$, of all the studied adsorption and desorption isotherms showed a non-linear trend, so that they were rather sigma shaped curves. The shape and the steadily decreasing trend of the generalized dimension, $D_q$, when $q$ moves from -5 to +5, and the ranking of the three first positive moments, i.e., $D_0, > D_1 > D_2$ suggests multifractal behavior of all NAIs and NDIs studied.

The entropy dimension, $D_1$, has been recognized as a measure of diversity and in our study case gauges the concentration degree of $N_2$ adsorption or desorption on a specific p/p$_0$ interval. When $D_1$ approaches $D_0$ ($D_0 = 1$), the measure is considered to be evenly distributed over all the scale measured, while $D_1$ values close to zero reflect the measure concentrates in a small size domain of scale (Halsey et al., 1986; Tarquis et al., 2006; Vidal Vázquez et al., 2008). Entropy or information dimension, $D_1$, of the 32 horizons studied (two repetitions per horizon) varied between 0.492 and 0.643, with a mean value





of 0.571, for adsorption isotherms, and between 0.620 and 0.797, with a mean value of 0.683, for desorption isotherms (Tables 3 and 4, and Table S2, as Supplementary content). Figure 5 shows the relationship between D1 values extracted from NAIs and NDIs. Mean values of $D_1$ for NAIs and NDIs were significantly different (P<0.05), as shown in Table S2. Lower values of $D_1$ for adsorption isotherms compared with desorption isotherms are consistent with previous work (Paz-Ferreiro et

al., 2013). The smaller the value of $D_1$ is, the higher the measure is concentrated in a small size domain of the studied scale. Both, nitrogen adsorption and desorption isotherms are sharper at the end of the curve (Figure 1), where the measure, in this case differential distribution of nitrogen volume for successive relative pressures, p/p$_0$, is subjected to rapid increases. However, adsorption changes by condensation is more abrupt than desorption changes by evaporation, because of the hysteresis loop. Hence, the measure is more evenly distributed for desorption than for adsorption isotherms.

The correlation dimension, $D_2$, showed a trend to decrease as $D_1$ decreased, although there were differences in the extent of the ($D_1 - D_2$) values, exhibiting various degrees of multifractality for adsorption and desorption isotherms.

Examples of f ($\alpha$)-$\alpha$ spectra for adsorption and desorption isotherms of medium and heavily textured soils are shown in Figure 6 and Figure 7, respectively. The singularity spectrum of all the studied nitrogen isotherms were concave down parabolic curves. Again, shape and asymmetry of the singularity spectra showed the scaling properties of NAs and NDIs

could be fitted reasonably well with multifractal models.

All the spectra were asymmetric, left-deviating curves, shorter toward the right and more or less longer toward the left; Thus, there were various degrees of asymmetry in the studied data sets. Asymmetry of the f ($\alpha$) spectrum toward the left indicates domination of high or presence of extremely high values in the probability distribution of the measure. Rare high events in $N_2$ differential adsorption and desorption were more frequent than rare low events. Hence, the general shape of the ($\alpha$)

spectra from adsorption and desorption isotherms is compatible with the rapid changes during $N_2$ condensation (at the adsorption phase) or evaporation (at the desorption phase) recorded for high relative pressures, i.e. , p/p$_0$ values approaching the unity.

The amplitude of the f($\alpha$) spectrum also is an indicator of heterogeneity, because it provides information on the diversity of he scaling exponents of a measure. So, the wider the f($\alpha$)-$\alpha$ spectrum is, the higher is the heterogeneity in the scaling indices.

Also the width of the generalized dimension spectra, which was assessed here by the difference w= ($D_{-5}$-$D_5$) can be considered as a measure of heterogeneity. Following these criteria, desorption isotherm demonstrated to be much more homogeneous than adsorption isotherms.

Values of Hölder exponent of order zero, $\alpha_0$, extracted form the singularity spectra of adsorption and desorption isotherms also are reported in Table 3 and 4, respectively. Parameter $\alpha_0$, quantifies the average degree of mass density of the measure.

The $\alpha_0$, values varied between 1.260 and 1.579 for adsorption isotherms and between 1.113 and 1.257 for desorption isotherms, with mean values of 1.477 and 1.206, respectively. These figures are relatively high and of the same order of magnitude as reported before for NAIs and NDIs (Paz-Ferreiro et al., 2009; Vidal-Vázquez and Paz-Ferreiro, 2012; Paz-Ferreiro et al., 2013). Opposite to entropy dimension, $D_1$, parameter $\alpha_0$, was higher for adsorption than for desorption isotherms. The relatively large values of exponent $\alpha_0$ and the smaller amplitude of NAI curves compared to NDI curves are

compatible with a higher heterogeneity and a lower anisotropy of the distribution of the measure for NAIs, compared to NDIs.

Summarizing, low $D_1$ values reflect the fact that most of the measure concentrates in a small size domain of the study scale, while high values of $D_1$ indicate that the measure is evenly distributed. Low $D_2$ means a small spatial autocorrelation and vice-versa. Moreover, large $\alpha_0$ and wide ($D_{-5}$-$D_5$) are characteristic of a high heterogeneous measure. Hence, adsorption



isotherms behaved as more clustered (i.e. less evenly distributed) measures, with lower entropy , $D_1$, and correlation, $D_2$, dimensions, higher heterogeneity and, in general, lower asymmetry, when compared with desorption isotherms. The multifractal parameters gave a good description of how the amount of $N_2$ gas rises and recedes in the absorption and desorption isotherms, respectively, in the scale range $0 < p/p_0 < 1$.

### 3.3 Texture effects on multifractal parameters from NAIs and NDI

Mean values of several multifractal parameters extracted from NAIs and NDIs are listed in Table 5, where one-way ANOVA analysis results are also shown. Parameters $D_{-5}$, $D_2$, $D_5$, $(D_{-5}-D_5)$ and $\alpha_0$, extracted from multifractal curves of NAIs were significantly different between the two contrasting groups of soils studied, while $D_1$ during absorption showed not significant differences ($P < 0.05$). On the other hand, parameters $D_{-5}$, $D_1$, $D_2$, $D_5$ from the generalized dimension function of desorption

isotherms showed also significant differences, $(D_{-5}-D_5)$ while $\alpha_0$ during desorption was not significantly different ($P < 0.05$) between these two soil groups.

Heterogeneity, given by parameter $(D_{-5}-D_5)$ was significantly greater for heavily textured, than for medium textured soils, over sandstone during adsorption ($P < 0.05$). Meanwhile during desorption there were no significant differences in mean values of $(D_{-5}-D_5)$, but these were slowly higher for soils over sandstone.

Parameters $D_{-5}$, $D_1$, $D_2$ and $D_5$, showed greater values for clayey soils during adsorption, but during desorption the trend was opposite and all of them were higher for the medium textured soils. This result suggest a more evenly distributed measure of the clayey soils  and medium textured during adsorption and desorption, respectively. These differences are consistent with the wider hysteresis loop of the medium textured soils compared to that of the clayey soils.

Hölder exponent of order 0 was higher for soils over weathered materials compared to those over sandstone, both for NAIs

and NDIs. However differences between these two soil groups were significant ($P < 0,05$) for adsorption isotherms, and not for desorption isotherms.

### 3.4 Multifractal parameters and general soil properties

Pearson product moment correlations between selected multifractal parameters ($D_{-5}$, $D_2$, $D_5$, $(D_{-5}-D_5)$ and $\alpha_0$) to clay content, SSA and organic carbon content are show in Table 6. This is not consistent wit the results of previous work (Paz-Ferreiro et

al:, 2013), which demonstrates that scaling heterogeneity showed a trend to increase as a function of clay content and to decrease as a function of organic carbon content, both  for NAIs and NDIs. Our results, however suggest that clay and organic carbon are factors that may determine the geometrical heterogeneity at the surface-pore interfaces of the studied soils in a different way with respect to previous studies. In other words, the nonlinearity of NAIs and NDIs of soils collected in Santa Catarina (Paz-Ferreiro et al; 2013) and in São Paulo) may be driven by different soil properties or processes. This

reinforces the need to further perform multifractal analysis of $N_2$ isotherms.

Principal component analysis (PCA) also was used to further asses the relationships between general soil properties and multifractal parameters. Results of PCA performed for two datasets, which included physicochemical properties and parameters resulting from multifractal analysis ($D_1$, $D_2$ and $\alpha_0$) of either absorption or desorption isotherms, are shown as Supplementary Digital Content (Table S2).

For the two data sets consisting of general soil properties and multifractal parameters from either NAIs or NDIs, the main contributions to the first axis were from pH, some properties of the exchange complex and sand content. So, pH, SB and V were best positively and sand content and exchangeable Al best negatively correlated to the scores of PC1, respectively ($r \geq$





|0.76|, P < 0.01). Other various soil properties were also correlated to PC1 scores, namely clay content, exchangeable H + Al and SSA, but showed a weaker correlation, meaning its contribution was much lesser.

The scores of the second axis were significantly (P > 0.01) and positively correlated to clay content, H+Al, CEC, V and SSA, while they exhibit negative correlations with silt and sand contents. Best correlated variables (r ≥ |0.76|, P < 0.01) were

SSA, silt and clay contents. Multifractal parameters assessed contributed or not to the second axis. So, for adsorption isotherms $D_{-5}$, $D_5$, $D_{-5} - D_5$, and $\alpha_0$ were positively correlated with the scores of PC2, but his was not the case for $D_1$ and $D_2$. However for desorption isotherms $D_1$ and $D_2$ showed stronger correlation with PC2 scores.

In the orthogonal space defined by PC1 and PC2, this second axes clearly separates profiles P1 to P3 from profiles P4 to P6 (Supplementary Digital Content Figure S2). Therefore, PCA showed soil surface properties, such as SSA obtained by

classical methods, and multifractal parameters were also useful to associate soil profiles with similar properties.

Realistic values of SSA have proven to be of great interest in several application related to soil environmental quality (Pachepsky et al., 1995; Petersen et al., 1996; Hajnos et al., 2000). The two studied soil groups from São Paulo State significantly differed in texture (clay, silt and san content), CEC and SSA. Sandy-loam and loamy soils with low SSA from undulated landscapes are most susceptible to clay dispersion, seal formation and heavy soil erosion. Clayey soils with

relatively high SSA from stable landscapes exhibit a high aggregate stability and infiltration rate and they are less susceptible to erosion. Also the former and more erodible soils are expected to exhibit high enrichment ratios for sediment, and associated nutrients and contaminants than the latter stable soils.

Mutifractal analysis is a powerful tool to describe the physical processes underlying nitrogen adsorption and desorption, and in this respect goes beyond parameters such as SSA, based on classical non fractal models. Thus, multifractal analysis offer

additional information of value as it reveals the hidden structure of adsorption and desorption isotherms. The choice of representing soil properties in terms of nonlinear process provide new insight for interpretation of the phenomena studied. In this perspective the information obtained could be useful for soil quality evaluations, based on properties and parameters that are inherent for deep soil horizons and affected by land use for top horizons at the soil surface.

In our study, multifractal analysis was used to evaluate $N_2$ isotherms from two contrasting soil groups. However, for a

horizon with a given texture, management system has been proven to influence SSA and multifractal characteristics of adsorption isotherms, as well (Paz Ferreiro et al., 2009). This suggests further analysis of $N_2$ adsorption and desorption isotherms from the topsoil horizon of a loamy textured or a clayey textured soil under different management systems could be useful for assessment of environmentally sound practices in the studied landscapes.

## 4. Conclusions

For all the collected samples, SSA showed a strong correlation with clay content. However, no significant relationship was found between these soil surface properties and properties of the soil exchangeable complex. SSA was significantly higher for clayey soils than for medium textured soils.

Nitrogen adsorption and desorption isotherms exhibited multifractal behaviour. However, NAIs were less evenly distributed measures than desorption isotherms, as indicated by lower entropy dimension, $D_1$. Also NDIs were more heterogeneous than

desorption isotherms, as the former exhibited higher widths of generalized dimension ($D_{-5} - D_5$) and singularity spectra($\alpha_{max}$-$\alpha_{min}$) than the later. Accordingly, multifractal parameters from adsorption and desorption isotherms were quite different. Contrasting multifractal behaviour of NAIs and NDIs proved to be mainly related to the characteristics of the hysteretic loop. Several other multifractal parameters extracted from NAIs and NDIs also were useful to distinguish between the medium





textured and clayey soils. This suggest that the nonlinearity of NAIs and NDIs of different soil types may be driven by different soil properties or processes.

There were significant correaltions between parameters $D_{-5}$ and ($D_{-5}$ - $D_5$) to clay content or SSA, which were positive and negative for NAIs and NDIs, respectively. Also for NDIs, parameters $D_{-5}$ and ($D_{-5}$ - $D_5$) were positively correlated to organic
carbon content. However, these relationships were no consistent with those found in previous work.

Altogether, multifractal analysis of NAIs and NDIs provided new information for describing the soil-pore interface in terms of nonlinear processes. This approach is considered as complementary to SSA determined by classical non-fractal approaches.

**Acknowledgements**. This work was partially supported by Spanish Ministry of Science and Technology (project CGL2013-
47814-C2) and by Xunta de Galicia (project 10MRU162037PR).

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

**Table 1.** General information about the studied samples: horizon, vertical limits, location, altitude, parent material, and classification following Brazilian System of Soil Classification (BSSC), Soil Survey Staff (SSS) and World Reference Base (WRB) for Soil Resources.

| Soil | Horizon | Depth/cm | Location | Parent Material | BSSC | SSS | WRB | Texture |
|---|---|---|---|---|---|---|---|---|
| 1 | Ap | 0-8 | Monte Mor (607 m) | Sandstone | Neossolo | *Typic* | Leptosol | loam |
| | C1 | 8-20 | 22°55'15.71''S | and silt | Regolítico | *Udorthent* | | loam |
| | C2 | 20-32 | 47°17'09,06''W | | | | | loam |
| 2 | Ap1 | 0-13 | Monte Mor (610 m) | Sandstone | Argissolo | *Typic* | Acrisol | loam |
| | Ap2 | 13-25 | 22°55'06.64''S | and silt | Amarelo | *Hapludult* | | sandy loam |
| | AB | 25-37 | 47°16'59.71''W | | Distrófico | | | sandy loam |
| | Bt | 37-54 | | | | | | sandy loam |
| | Bt/Cr1 | 54-78 | | | | | | loam |
| 3 | Ap1 | 0-15 | Monte Mor (622m) | Sandstone | Argissolo | *Arenic* | Acrisol | sandy loam |
| | Ap2 | 15-30 | 22°54'26.97''S | and silt | Vermelho | *Hapludult* | | sandy loam |
| | A2/E | 30-42 | 47°17'12.78''W | | Amarelo | | | sandy loam |
| | E | 42-62 | | | Distrófico | | | sandy loam |
| | Bt | 62-92 | | | | | | loam |
| | Bt/Cr | +92 | | | | | | loam |
| 4 | Ap1 | 0-20 | Sumaré (640 m) | Weathered material | Latossolo | *Humic* | Ferralsol | loam |
| | Ap2 | 20-40 | 22°52'21.04''S | from basic rocks | Vermelho | *Hapludox* | | sandy clay |
| | A21 | 40-70 | 47°18'18.69''W | (Diabase) | Amarelo | | | clay |
| | A22 | 70-100 | | | Distrófico | | | clay |
| | A23 | 100-130 | | | húmico | | | clay |
| | A24 | 130-150 | | | | | | clay |
| | A25 | 150-180 | | | | | | clay |
| | Bw1 | 250-300 | | | | | | clay |
| 5 | Ap | 0-10 | Sumaré (574 m) | Weathered material | Nitossolo | *Typic* | Nitisol | clay |
| | B1 | 10-35 | 22°47'33.58''S | from clayey and | Vermelho | *Rhodudult* | | clay |
| | B21 | 35-60 | 47°20'1.64''W | loamy sediments | Distroférrico | | | clay |


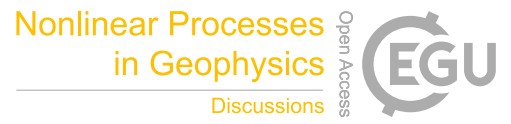

|   |     |         |         |                    |                | *típico*            |           | clay |
|---|-----|---------|---------|--------------------|----------------|---------------------|-----------|------|
|   | B22 | 60-76   |         |                    |                |                     |           | clay |
|   | B23 | 76-104  |         |                    |                |                     |           | clay |
| 6 | Ap  | 0-18    | Campinas (620 m) | Weathered material | Latossolo | *Rhodic* | Ferralsol | clay |
|   | AB  | 18-36   | 22°49'11''S | from basic rocks | Vermelho | *Hapludox* |        | clay |
|   | Bw1 | 36-73   | 47°03'43''W | (Diabase) | Distroférrico |        |          | clay |
|   | Bw2 | 73-117  |         |                    | *típico*       |                     |           | clay |
|   | Bw3 | 117-158 |         |                    |                |                     |           | clay |

**Table 2.** Correlation matrix for sand, silt, clay, organic matter content (OM), pH, complex exchange properties and surface properties

|        | Sand      | Silt      | Clay     | OM       | pH       | H + Al   | SB      | CEC      | V      | SSA |
|--------|-----------|-----------|----------|----------|----------|----------|---------|----------|--------|-----|
| **Sand** | 1       |           |          |          |          |          |         |          |        |     |
| **Silt** | 0.406*  | 1         |          |          |          |          |         |          |        |     |
| **Clay** | -0.892** | -0.775** | 1        |          |          |          |         |          |        |     |
| **OM**   | -0.235  | 0.077     | 0.124    | 1        |          |          |         |          |        |     |
| **pH**   | -0.689** | 0.045    | 0.454**  | 0.032    | 1        |          |         |          |        |     |
| **H + Al** | 0.113 | -0.529** | 0.183    | 0.322    | -0.622** | 1        |         |          |        |     |
| **SB**   | -0.557** | 0.280    | 0.247    | 0.481**  | 0.700**  | -0.464** | 1       |          |        |     |
| **CEC**  | -0.098  | -0.480** | 0.305    | 0.555*** | -0.415*  | 0.934**  | -0.116  | 1        |        |     |
| **V**    | -0.438* | 0.404*    | 0.103    | 0.110    | 0.835**  | -0.783** | 0.866*  | -0.529** | 1      |     |
| **SSA**  | -0.878*** | -0.779*** | 0.992** | 0.087  | 0.436*   | 0.185    | 0.239   | 0.303    | 0.101  | 1   |

(* and **, correspond to $P < 0.05$ and $P < 0.01$, respectively)

Abbreviations: (H+Al = exchangeable H + Al, SB = sum of exchangeable base, i.e., K+ Mg+ Ca, CEC = cation exchange capacity, V = percent base saturation, SSA = specific surface area and $V_{0.95}$= cumulative $N_2$ volume adsorbed at 0.95 relative pressure)



**Table 3.** Multifractal parameters extracted from the generalized dimension function ($D_{-5}$, $D_1$, $D_2$, $D_5$) and from the singularity spectrum ($\alpha_0$) of nitrogen adsorption isotherms.(NAIs).

| Horizon/depth | $D_{-5}$ | $D_1$ | $D_2$ | $D_5$ | $\alpha_0$ |
|---|---|---|---|---|---|
| **Typic Udorthent** | | | | | |
| Ap (0-8) | 2.309 ± 0.346 | 0.521 ± 0.036 | 0.343 ± 0.035 | 0.229 ± 0.022 | 1.500 ± 0.016 |
| C1 (8-20) | 1.397 ± 0.170 | 0.568 ± 0.028 | 0.397 ± 0.028 | 0.267 ± 0.020 | 1.320 ± 0.027 |
| C2 (20-32) | 2.366 ± 0.334 | 0.513 ± 0.018 | 0.390 ± 0.020 | 0.261 ± 0.015 | 1.509 ± 0.029 |
| **Typic Hapludult** | | | | | |
| Ap1 (0-13) | 1.999 ± 0.289 | 0.568 ± 0.043 | 0.403 ± 0.045 | 0.276 ± 0.033 | 1.439 ± 0.019 |
| Ap2 (13-25) | 1.168 ± 0.122 | 0.615 ± 0.037 | 0.461 ± 0.043 | 0.319 ± 0.033 | 1.266 ± 0.027 |
| AB (25-37) | 1.168 ± 0.118 | 0.621 ± 0.034 | 0.469 ± 0.040 | 0.325 ± 0.031 | 1.260 ± 0.025 |
| Bt (37-54) | 1.517 ± 0.262 | 0.642 ± 0.042 | 0.534 ± 0.061 | 0.388 ± 0.052 | 1.326 ± 0.027 |
| Bt/Cr1 (54-78) | 1.274 ± 0.203 | 0.643 ± 0.035 | 0.523 ± 0.050 | 0.373 ± 0.041 | 1.280 ± 0.021 |
| **Arenic Hapludult** | | | | | |
| Ap1 (0-15) | 2.051 ± 0.310 | 0.505 ± 0.038 | 0.313 ± 0.032 | 0.208 ± 0.022 | 1.462 ± 0.023 |
| Ap2 (15-30) | 1.547 ± 0.181 | 0.523 ± 0.044 | 0.349 ± 0.045 | 0.235 ± 0.029 | 1.916 ± 0.017 |
| A2/ E (30-42) | 2.357 ± 0.379 | 0.531 ± 0.043 | 0.360 ± 0.042 | 0.243± 0.030 | 1.522 ± 0.021 |
| E (42-60) | 2.169 ± 0.338 | 0.524 ± 0.041 | 0.356 ± 0.041 | 0.240 ± 0.029 | 1.495 ± 0.020 |
| Bt (62-92) | 1.148 ± 0.123 | 0.596 ± 0.044 | 0.427 ± 0.048 | 0.294 ± 0.036 | 1.278± 0.032 |
| Bt/Cr (>92) | 1.421 ± 0.209 | 0.548 ± 0.041 | 0.374 ± 0.042 | 0.253 ± 0.031 | 1.357± 0.031 |
| **Mean group 1** | **1.986** | **0.566** | **0.407** | **0.279** | **1.424** |
| | | | | | |
| **Humic Hapludox** | | | | | |
| Ap1 (0-20) | 2.710 ± 0.378 | 0.579 ± 0.071 | 0.461 ± 0.092 | 0.345 ± 0.081 | 1.547± 0.016 |
| Ap2 (20-40) | 2.581 ± 0.364 | 0.609 ± 0.074 | 0.512 ± 0.110 | 0.410 ±  0.111 | 1.518 ± 0.025 |
| A21 (40-70) | 2.261 ± 0.314 | 0.597 ± 0.066 | 0.482 ± 0.092 | 0.366 ± 0.085 | 1.478 ± 0.028 |
| A22 (70-100) | 2.385 ± 0.334 | 0.559 ± 0.060 | 0.420 ± 0.072 | 0.298 ±  0.057 | 1.515 ± 0.026 |
| A23 (100-130) | 2.642 ± 0.381 | 0.585 ± 0.076 | 0.475 ± 0.105 | 0.367 ± 0.098 | 1.534 ± 0.030 |
| A24 (130-150) | 2.847 ± 0.423 | 0.567 ± 0.069 | 0.440 ± 0.087 | 0.325 ± 0.075 | 1.571 ± 0.026 |
| A25 (150-180) | 2.904 ± 0.463 | 0.554 ± 0.063 | 0.412 ± 0.073 | 0.292 ± 0.058 | 1.568 ± 0.028 |
| Bw1 (250-300) | 2.566 ± 0.430 | 0.565 ± 0.063 | 0.423 ± 0.075 | 0.301 ± 0.060 | 1.518 ± 0.038 |
| **Typic Rhodudult** | | | | | |
| Ap (0-10) | 2.514 ± 0.423 | 0.580 ± 0.057 | 0.441 ± 0.069 | 0.314 ± 0.056 | 1.511 ± 0.033 |
| B1 (10-35) | 2.346 ± 0.364 | 0.562 ± 0.057 | 0.413 ± 0.066 | 0.290 ± 0.052 | 1.485 ± 0.031 |
| B21 (35-60) | 2.556 ± 0.392 | 0.565 ± 0.059 | 0.419 ± 0.068 | 0.295 ± 0.053 | 1.514 ± 0.027 |
| B22 (60-76) | 2.632± 0.415 | 0.578 ± 0.059 | 0.439 ± 0.071 | 0.313 ± 0.057 | 1.505 ± 0.031 |
| B23 (76-104) | 2.914 ± 0.514 | 0.529 ± 0.061 | 0.376 ± 0.069 | 0.265 ± 0.054 | 1.578 ± 0.037 |
| **Rhodic Hapludox** | | | | | |
| Ap (0-18) | 2.418 ± 0.384 | 0.602 ± 0.064 | 0.482 ± 0.084 | 0.356 ± 0.072 | 1.484 ± 0.030 |
| AB (18-36) | 2.402 ± 0.360 | 0.579 ± 0.061 | 0.442 ± 0.073 | 0.316 ± 0.059 | 1.497 ± 0.027 |
| Bw1 (36-73) | 2.670 ± 0.415 | 0.593 ± 0.068 | 0.474 ± 0.091 | 0.358 ± 0.083 | 1.513 ± 0.032 |
| Bw2 (73-117) | 2.438 ± 0.394 | 0.578 ± 0.065 | 0.444 ± 0.080 | 0.321 ± 0.067 | 1.494 ± 0.035 |
| Bw3 (117-158) | 2.557 ± 0.423 | 0.580 ± 0.069 | 0.455 ± 0.090 | 0.339 ± 0.079 | 1.508 ± 0.036 |
| **Mean group 2** | **1.998** | **0.576** | **0.445** | **0.288** | **1.519** |



**Table 4.** Multifractal parameters extracted from the generalized dimension function ($D_{-5}$, $D_1$, $D_2$, $D_5$) and from the singularity spectrum ($\alpha_0$) of nitrogen desorption isotherms.(NDIs).

| Horizon/depth | $D_{-5}$ | $D_1$ | $D_2$ | $D_5$ | $\alpha_0$ |
|---|---|---|---|---|---|
| **Typic Udorthent** | | | | | |
| Ap (0-8) | 1.546 ± 0.065 | 0.678 ± 0.013 | 0.528 ± 0.027 | 0.370 ± 0.027 | 1.257 ± 0.002 |
| C1 (8-20) | 1.401 ± 0.044 | 0.708 ± 0.014 | 0.613 ± 0.031 | 0.451 ± 0.029 | 1.204 ± 0.009 |
| C2 (20-32) | 1.388 ± 0.056 | 0.675 ± 0.014 | 0.610 ± 0.026 | 0.435 ± 0.035 | 1.198 ± 0.013 |
| **Typic Hapludult** | | | | | |
| Ap1 (0-13) | 1.755± 0.169 | 0.720 ± 0.022 | 0.628 ± 0.047 | 0.474 ± 0.050 | 1.257 ± 0.004 |
| Ap2 (13-25) | 1.257 ± 0.046 | 0.757 ± 0.034 | 0.683 ± 0.066 | 0.538 ± 0.078 | 1.148 ± 0.017 |
| AB (25-37) | 1.256 ± 0.049 | 0.762 ± 0.031 | 0.696 ± 0.063 | 0.551 ± 0.077 | 1.142 ± 0.015 |
| Bt (37-54) | 1.187 ± 0.039 | 0.797 ± 0.035 | 0.769 ± 0.073 | 0.665 ± 0.101 | 1.115 ± 0.016 |
| Bt/Cr1 (54-78) | 1.216 ± 0.052 | 0.797 ± 0.031 | 0.786 ± 0.067 | 0.686 ± 0.096 | 1.114 ± 0.014 |
| **Arenic Hapludult** | | | | | |
| Ap1 (0-15) | 1.245 ± 0.054 | 0.620 ± 0.053 | 0.439 ± 0.069 | 0.310 ± 0.058 | 1.241 ± 0.033 |
| Ap2 (15-30) | 1.301 ± 0.067 | 0.640 ± 0.053 | 0.483 ± 0.075 | 0.349 ± 0.066 | 1.242 ± 0.028 |
| A2/ E (30-42) | 1.277 ± 0.068 | 0.648 ± 0.054 | 0.496 ± 0.077 | 0.362 ± 0.069 | 1.224 ± 0.029 |
| E (42-60) | 1.218 ± 0.061 | 0.660 ± 0.048 | 0.503 ± 0.069 | 0.362 ± 0.061 | 1.212 ± 0.029 |
| Bt (62-92) | 1.174 ± 0.051 | 0.723 ± 0.059 | 0.625 ± 0.102 | 0.524 ± 0.123 | 1.172 ± 0.031 |
| Bt/Cr (>92) | 1.227 ± 0.053 | 0.678 ± 0.058 | 0.554 ± 0.094 | 0.431 ± 0.097 | 1.203 ± 0.031 |
| **Mean group 1** | **1.318** | **0.704** | **0.601** | **0.465** | **1.195** |
| | | | | | |
| **Humic Hapludox** | | | | | |
| Ap1 (0-20) | 0.851 ± 0.032 | 0.653 ± 0.048 | 0.494 ± 0.059 | 0.350 ± 0.048 | 1.224 ± 0.033 |
| Ap2 (20-40) | 0.850 ± 0.036 | 0.664 ± 0.039 | 0.496 ± 0.046 | 0.347 ± 0.033 | 1.213 ± 0.029 |
| A21 (40-70) | 0.872 ± 0.036 | 0.650 ± 0.038 | 0.474 ± 0.043 | 0.327± 0.036 | 1.217 ± 0.029 |
| A22 (70-100) | 0.793 ± 0.033 | 0.666 ± 0.057 | 0.526 ± 0.072 | 0.399 ± 0.080 | 1.220 ± 0.035 |
| A23 (100-130) | 0.858 ± 0.035 | 0.638 ± 0.047 | 0.468 ± 0.056 | 0.327 ± 0.064 | 1.228 ± 0.034 |
| A24 (130-150) | 0.840 ± 0.035 | 0.644 ± 0.050 | 0.480 ± 0.062 | 0.340 ± 0.070 | 1.228 ± 0.035 |
| A25 (150-180) | 0.851 ± 0.039 | 0.642 ± 0.051 | 0.479 ± 0.054 | 0.340 ± 0.052 | 1.229 ± 0.034 |
| Bw1 (250-300) | 0.785 ± 0.031 | 0.666 ± 0.059 | 0.516 ± 0.064 | 0.380 ± 0.072 | 1.217 ± 0.036 |
| **Typic Rhodudult** | | | | | |
| Ap (0-10) | 0.705 ± 0.040 | 0.709 ± 0.061 | 0.602 ± 0.098 | 0.485 ± 0.105 | 1.194 ± 0.032 |
| B1 (10-35) | 0.756 ± 0.033 | 0.678 ± 0.063 | 0.544 ± 0.083 | 0.418 ± 0.091 | 1.207 ± 0.035 |
| B21 (35-60) | 0.793 ± 0.040 | 0.670 ± 0.059 | 0.529 ± 0.065 | 0.396 ± 0.079 | 1.212 ± 0.034 |
| B22 (60-76) | 0.794 ± 0.037 | 0.676 ± 0.050 | 0.525 ± 0.055 | 0.378 ± 0.065 | 1.201 ± 0.032 |
| B23 (76-104) | 0.695 ± 0.037 | 0.661 ± 0.077 | 0.542 ± 0.078 | 0.468 ±0.135 | 1.226 ± 0.041 |
| **Rhodic Hapludox** | | | | | |
| Ap (0-18) | 0.807 ± 0.039 | 0.680 ± 0.045 | 0.531 ± 0.049 | 0.384 ± 0.051 | 1.202 ± 0.029 |
| AB (18-36) | 0.737 ± 0.031 | 0.691 ± 0.061 | 0.570 ± 0.092 | 0.445 ± 0.072 | 1.202 ± 0.034 |
| Bw1 (36-73) | 0.756 ± 0.039 | 0.687 ± 0.054 | 0.554 ± 0.077 | 0.420 ± 0.093 | 1.202 ± 0.033 |
| Bw2 (73-117) | 0.830 ± 0.037 | 0.656 ± 0.047 | 0.488 ± 0.049 | 0.343 ± 0.053 | 1.212 ± 0.033 |
| Bw3 (117-158) | 0.834 ± 0.037 | 0.650 ± 0.051 | 0.488 ± 0.063 | 0.347 ± 0.071 | 1.217 ± 0.034 |
| **Mean group 2** | **1.183** | **0.666** | **0.517** | **0.383** | **1.214** |





**Table 5**. Mean values of multifractal parameters from $N_2$ adsorption and desorption isotherms for the two studied soil groups, and results of one way ANOVA analysis.

| Texture | $D_{-5}$ | $D_1$ | $D_2$ | $D_5$ | $(D_{-5} - D_5)$ | $\alpha_0$ |
|---|---|---|---|---|---|---|
| | Adsorption (NAI) | | | | | |
| Medium | 1.986 | 0.563 | 0.405 | 0.279 | 1.706 | 1.396 |
| Clayey | 2.575 | 0.576 | 0.445 | 0.326 | 2.249 | 1.519 |
| F value | 12.598 | 0.857 | 4.648 | 8.645 | 19.148 | 21.460 |
| p* | **0.001** | 0.362 | **0.039** | **0.006** | **0.000** | **0.000** |
| | | | | | | |
| | Desorption (NDI) | | | | | |
| Medium | 1.318 | 0.704 | 0.601 | 0.465 | 0.853 | 1.195 |
| Clayey | 1.183 | 0.666 | 0.517 | 0.383 | 0.800 | 1.214 |
| F value | 25.762 | 7.361 | 9.731 | 7.377 | 0.966 | 2.528 |
| p* | **0.000** | **0.011** | **0.004** | **0.011** | 0.334 | 0.122 |

(Medium textured are horizons of soil profiles P1, P2 and P3; Clayey textured are horizons of soil profiles P4.P5 and P6; p* bold indicate that the results are significantly different.)

**Table 6.** Correlation coefficients among multifractal parameters obtained from $D_q$-q ($D_{-5}$, $D_1$, $D_2$, $D_5$, $D_{-5}$-$D_5$), and f($\alpha$)-$\alpha$ ($\alpha_0$)functions and general soil properties (clay, SSA and organic carbon)

| | $D_{-5}$ | $D_1$ | $D_2$ | $D_5$ | $(D_{-5}-D_5)$ | $\alpha_0$ |
|---|---|---|---|---|---|---|
| | Adsorption | | | | | |
| Clay percent | 0.597** | 0.131 | 0.315 | 0.406* | 0.549** | 0.315 |
| SSA | 0.599** | 0.157 | 0.357 | 0.444* | 0.546** | 0.302 |
| Organic carbon | 0.276 | -0.098 | -0.105 | 0.030 | 0.274 | 0.135 |
| | Desorption | | | | | |
| Clay percent | -0.531** | -0.365 | -0.405* | -0.346 | -0.531** | -0.212 |
| SSA | -0.569** | -0.349 | -0.380 | -0.324 | -0.569** | -0.161 |
| Organic carbon | 0.568** | -0.124 | -0.145 | -0.249 | 0.568** | 0.442* |







**Figure 1.** Examples of Nitrogen adsorption-desorption isotherms for samples from two horizons with contrasting texture (Profile 1, horizon Ap and Profile 6 , horizon Ap).

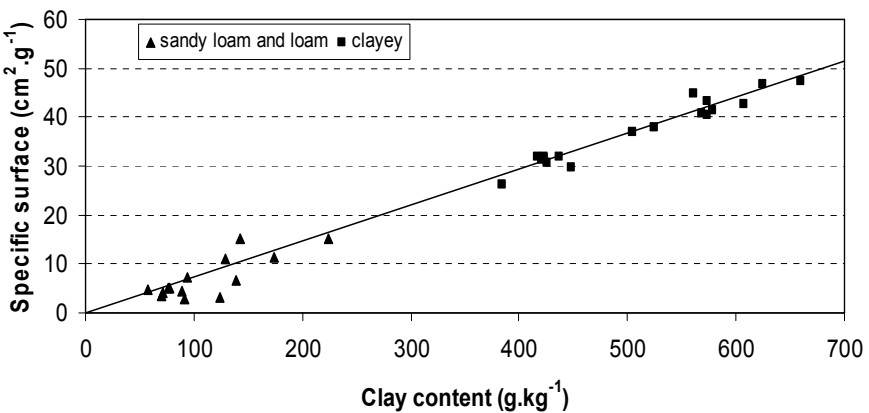

**Figure 2.** Relationship between clay content and SSA for all the horizons of horizons studied.

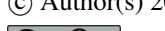

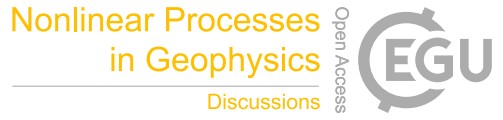

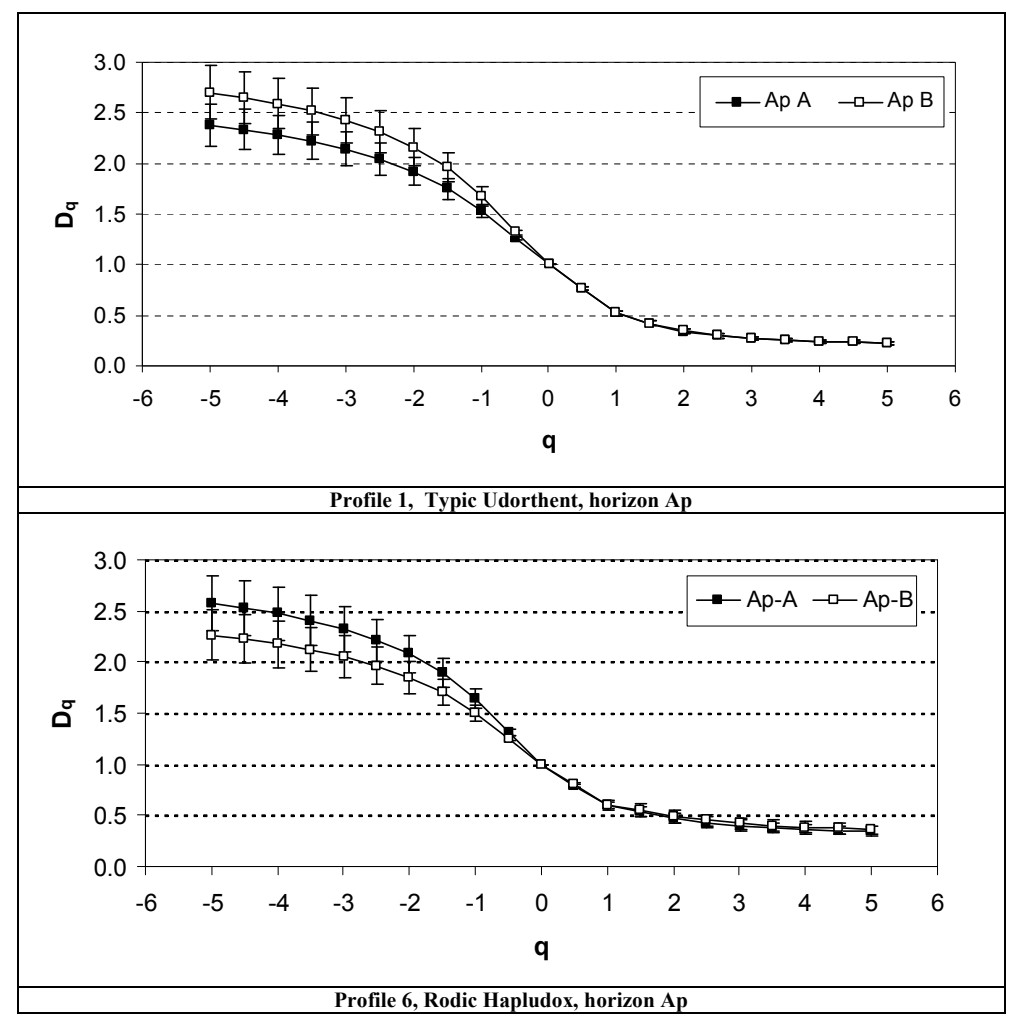

**Figure 3.** Selected examples of singularity spectra for adsorption isotherms (NAIs) of soil horizons with contrasting texture (Profile 1, horizons Ap and C1 and profile 6, horizons Ap and AB). Captions A and B in the legend indicate two repetitions per sample.





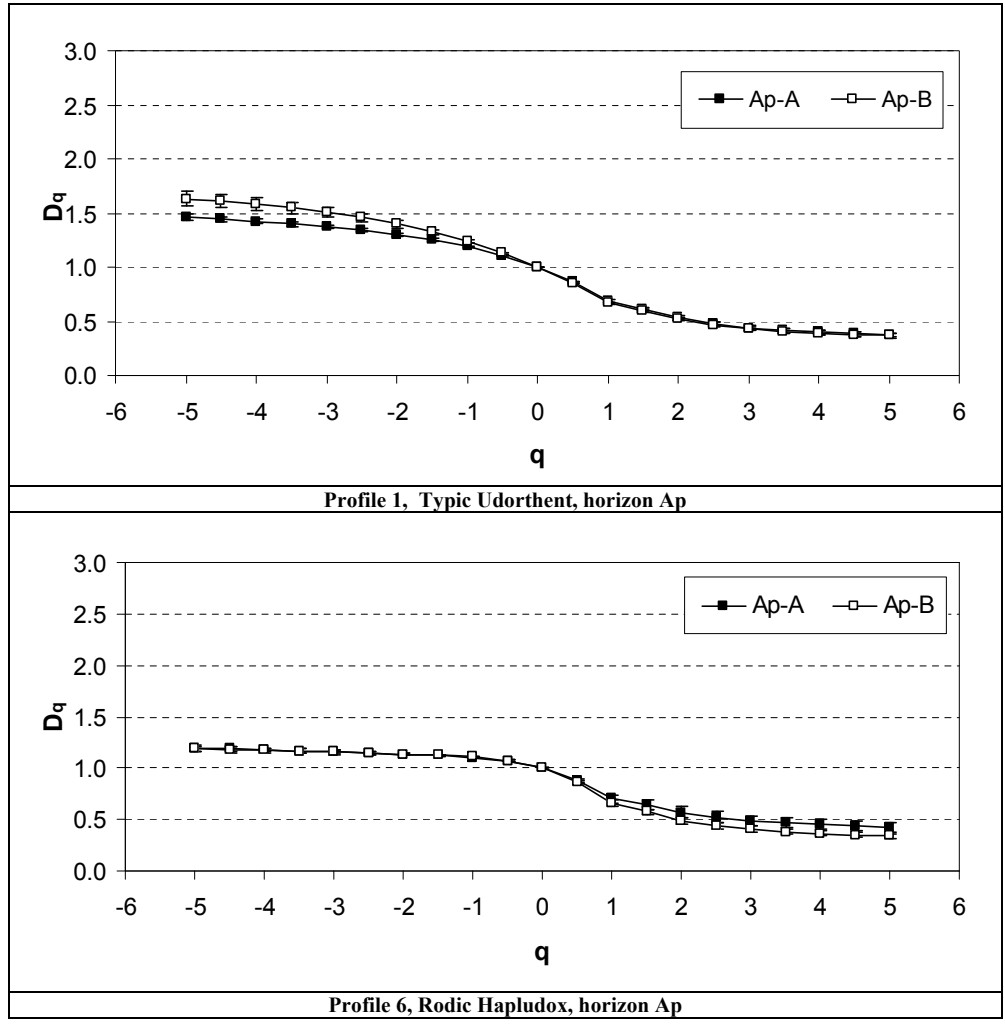

**Figure 4.** Selected examples of singularity spectra for desorption isotherms (NDIs) of soil horizons with contrasting texture (Profile 1, horizons Ap and C1 and profile 6, horizons Ap and AB). Captions A and B in the legend indicate two repetitions per sample.





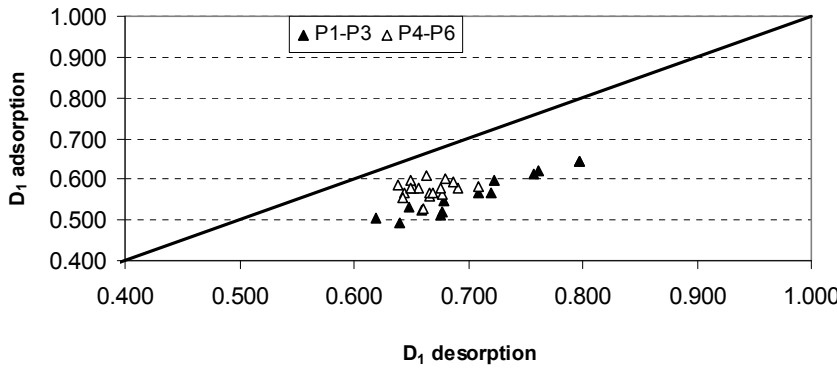

**Figure 5.** Relationships between entropy dimension, $D_1$, from $N_2$ adsorption and desorption isotherms. (P1-P3 = soils over sand- and siltstones poor in bases, P4-P6 = soils over weathered allochthonous parent material).





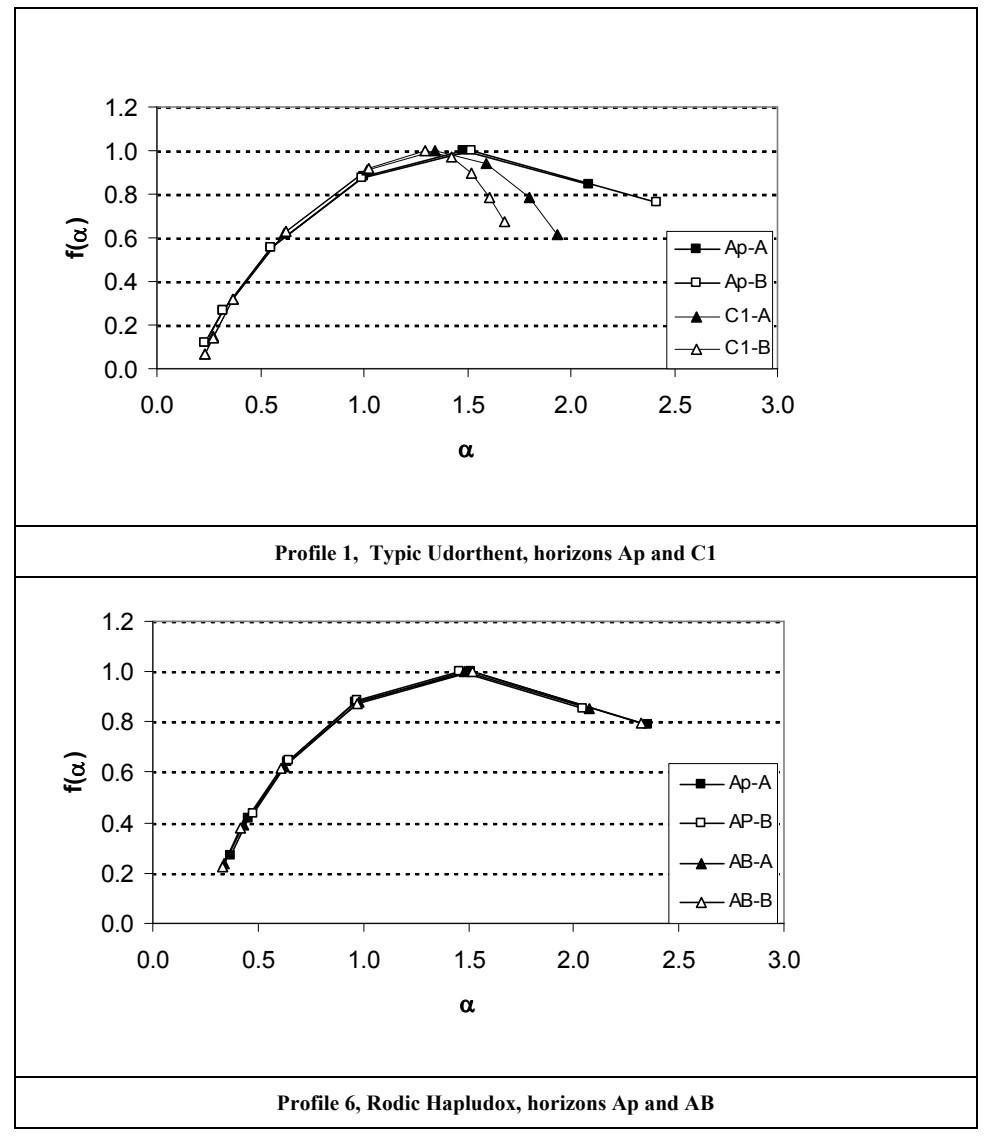

**Figure 6**. Selected examples of generalized dimension spectra for adsorption isotherms of soil horizons with contrasting texture (Profile 1, horizon Ap and profile 6, horizon Ap ). Captions A and B in the legend indicate two repetitions per sample.





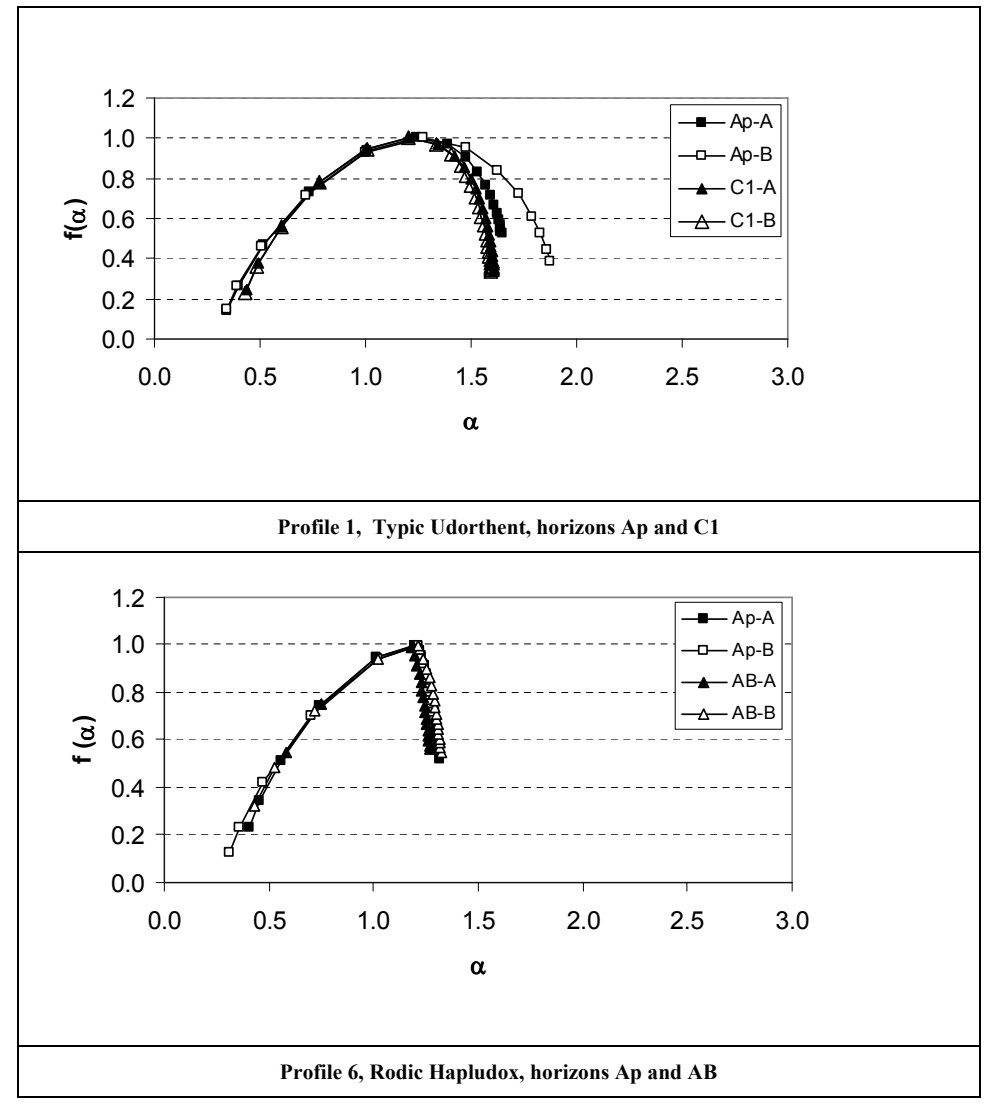

**Profile 1, Typic Udorthent, horizons Ap and C1**

**Profile 6, Rodic Hapludox, horizons Ap and AB**

**Figure 7**. Selected examples of generalized dimension spectra for desorption isotherms of soil horizons with contrasting texture (Profile 1, horizon Ap and profile 6, horizon Ap ). Captions A and B in the legend indicate two repetitions per sample.