# Peer review of "Multiscale analysis of nitrogen adsorption and desorption isotherms in soils with contrasting pedogenesis and texture"

_Nonlinear Processes in Geophysics, 2015_

## Referee Comment (RC2) · Anonymous Referee #2 · 29 Feb 2016

**Review of npg-2015-79**

This manuscript presents multifractal analyses of nitrogen adsorption and desorption isotherms obtained on soil samples from 6 different soil profiles in São Paulo State, Brazil. There is very little that is new here, and the manuscript essentially repeats the study by Paz-Ferreiro et al. (2013), but in a different geographic area and with some less clayey soils. Although it lacks originality, I suppose there is some incremental new knowledge gained on the effects of soil texture on these types of multifractal analyses. Therefore, I recommend acceptance following minor revisions. My specific comments are itemized below (page #, line #):

1,12 & 4,27 – If SSA implies Euclidean geometry & method of moments analyses indicate multifractal geometry is it valid to present both sets of results for the same samples. Surely, the soil pore space is either Euclidean or multifractal, but not both. I am not sure that correlating SSA with generalized dimensions and Hölder exponents, as is done in Table 6, is a useful exercise.

1,18 – If the nitrogen adsorption and desorption isotherms are indeed multifractal, what are the implications for soil pore size distributions? We have well-established physical models for Euclidean (e.g., random packing of uniform spheres) and monofractal (e.g., randomized Menger sponge) soils. Physically, what would a multifractal model for soil pore space geometry look like?

2,3 – Artificial is spelled incorrectly.

2,31 – Paz-Ferreiro and Vidal Vázquez (2012) does not appear in the references.

5,1 & Fig. 1 – It would be preferable to show the differential results rather than the cumulative plots in Fig. 1, which are very smooth and give absolutely no indication of multifractality.

6,5 – How many data points are needed to perform a robust multifractal analysis? Are 41 to 52 data points acceptable, and if so, according to what criteria?

6,6 – Report $R^2$ values for the log-log linear regression analyses. Also, the residuals should be examined for the absence of trend.

6,7 – For a true multifractal the normalized measure versus scale is always linear on a log-log scale regardless of the subdivision level (k). Does the observation that these plots were non-linear for k < 1 imply that this is a pre-multifractal system operating over a limited range of scales, or is it due to the limited number of data points used in the analyses?

6,7 – Why was the range $-5 \leq q \leq +5$ chosen? Did the linearity of the plots change with $q$, and if so what are the implications of this? Again, for a true multifractal, linearity should not be a function of $q$.

---

## Referee Comment (RC3) · Anonymous Referee #3 · 16 Mar 2016

The multifractal analyses of nitrogen adsorption and desorption isotherms is obtained from soil samples of 6 different profiles in São Paulo State, Brazil and the effect of soil texture on the multifractal patterns of the isotherms is studied. A similar paper wrote by Paz-Ferreiro et al. (2013) presented a similar analysis with a different database from a different experimental site.

In my opinion the basis of the analysis that has been performed has serious problems, even if the potential value of the conclusions of the analysis in the context of soil sciences is high. This is because the analysis itself is not well founded from a numerical point of view. Specifically, it is not clear that any conclusion about the multifractal behavior of a one dimensional series (temporal, spatial or of any other nature) with 41

or 52 data points could be reliable. In that respect I would put forward the work of A. Turiel, C. J. Pérez-Vicente and J. Grazzini entitle "Numerical methods for the estimation of multifractal singularity spectra on sampled data: A comparative study" (Journal of Computational Physics 216 (2006) 362–390). They study appropriate methods to assess multifractality over experimental discretized data and establish criteria to measure the confidence degree on the estimates. Manuscript npg 2105-72 do not comply with that criteria.

I understand that a mayor revision of the work should be undertaken in order to adapt the length of the series the authors use in their investigation to the type of analysis they propose. It is imperative to insure the consistency of the work done taking into account the potential benefit of the implications of this type of analysis for the "soil community".

---

## Referee Comment (RC4) · M. G. Wilson (Referee) · 1 Apr 2016

The manuscript should be accepted as is. The length of the manuscript is somewhat excessive. The Table 2 is not needed.

---

## Editor Comment (EC1) · A. Biswas (Editor) · 27 Apr 2016

Dear Authors, I am glad to inform that I have received comments form 4 reviewers on your manuscript (npg-2015-79). Most of the reviewers had a good saying about the manuscript. However, questions came up and needs to be addressed before moving to next step. Specifically, a question came up about the novelty of the manuscript and it's difference from the previous manuscript. I think you need to mention why and how this work is different from previous work(s) and its novel contribution. Secondly, few fundamental issues were raised by the reviewers (specifically #2 and #3) about the methodology itself. So, I would request authors to address those comments very carefully. Reviewers also mentioned about the length of the manuscript. Reduction

in length by tightening the content would make the manuscript a better read. Some more detailed comments and edits are provided by the reviewers. I would request to address the comments carefully and modify the manuscript accordingly. Based on the comments, I recommend a moderate revision prior to publication in the special issue. Thank you very much.

---

## Author Comment (AC1) · 16 Jul 2016

(1) Comments from Referee # 4 () The manuscript should be accepted as is. The length of the manuscript is somewhat excessive. The Table 2 is not needed.

(2) Author's response Thank you very much for your general assessment of this manuscript. Table 2 shows the correlation matrix between soils specific surface area (SSA) and the main general soil properties of the soil. This issue has been addressed in the Results and Discussion section, under subsection 3.1. We moved this Table to the Supplemental Digital Content.

(3) Author's changes in manuscript Table 2 from the initial manuscript appears now as Supplemental Digital Content (Table S2). Accordingly, subsection 3.1. was modified to indicate where the Table can be found.

Please also note the supplement to this comment:
http://www.nonlin-processes-geophys-discuss.net/npg-2015-79/npg-2015-79-AC1-supplement.pdf
* * *

---

## Author Comment (AC2) · 17 Jul 2016

(1) Comments from Referee # 4 () See annotated PDF. For tables 3 and 4, make a summary table in the main paper, and send the complete tables to digital supplements. Figure S2 (referred now as Fig S3 intext) must be placed in main text as Figure 6.

(2) author's response Thank you very much for your valuable comments. Specially, thank you very much for your patience in helping us with your detailed comment and correction of English language and typing mistakes provided in an annotated version of the manuscript. All your remarks have been helpful and have been taken into account. On the other hand, Tables 3 and 4 of the initial manuscript have been moved to the

Supplementary Digital Contents. (These Tables have not been summarized, as suggested, because we consider that Table 3 in the revised version, former Table S2, is a succinct summary). Indeed, Table S2 has been moved from the Supplementary Digital Content to the revised text, now as Table 2. Revision changes following your review have been marked in an annotated manuscript, which is sent as an attachement.

(3) author's changes in manuscript Tables 3 and 4 from the initial manuscript appears in the revised version as Supplemental Digital Content (Table S3 and S4, respectively). Table S2 from the initial Supplementary Digital Content appears now as Table 3 of the revised manuscript. Changes in the manuscript, due to typing error and language mistakes can be observed in an annotated version.

Please also note the supplement to this comment:
http://www.nonlin-processes-geophys-discuss.net/npg-2015-79/npg-2015-79-AC2-supplement.pdf

**Supplement:**

**Multiscale analysis of nitrogen adsorption and desorption isotherms in soils with contrasting pedogenesis and texture**

Jorge Paz-Ferreiro[1], Mara de A. Marinho[2], Cleide A. de Abreu[3], Eva Vidal-Vázquez[4]

[1]Royal Melbourne Institute of Technology University, School of Civil, Environmental and Chemical Engineering, Melbourne, Australia.

[2]Faculdade de Engenharia Agricola (FEAGRI), Universidade Estadual de Campinas (UNICAMP), Av. Candido Rondon, 501, Campinas, 13083-875, SP, Brazil.

[3]Instituto Agronômico de Campinas (IAC), Av. Barão de Itapura, 1481, Campinas, 13020-902, SP, Brazil.

[4]Facultad de Ciencias, Universidade da Coruña, Campus de Elviña, sn. Coruña, Spain.

*Correspondence to*: jpaz@udc.es

**Abstract.** The specific surface area (SSA) of a soil is commonly estimated from adsorption isotherms determined in a limited range of relative pressures ($p/p_0$), admitting a non fractal model. Nitrogen adsorption (NAI) and desorption (NDI) isotherms determined over the full range of $p/p_0$ have been described using the multifractal approach. This study aimed to assess effects of soil texture on the multifractality of NAIs and NDIs, and to analyze the association between multifractal parameters and soil properties. Six soil profiles were taken to get two groups of samples with contrasting pedogenetic origin, texture (medium or clayey), susceptibility to water erosion and quality for agricultural uses. These two soil groups also were significant differences in SSA and cation exchange capacity (CEC), but not in organic matter content (OMC). Consistent with previous studies, the scaling properties of both NAIs and NDIs from all the soil horizons studied could be fitted reasonably well with multifractal models. Values of parameters $D_{-5}$, $D_1$, $D_2$ and $D_5$, extracted from the generalized dimension function, $D_q$, were higher for clayey soils during adsorption, but during desorption all of them were higher for medium textured soils. Therefore, the measure was more evenly distributed for clayey soils during adsorption and for medium textured soils during desorption. Width of $D_q$ function given by parameter ($D_{-5}$-$D_5$) was significantly higher in clayey soils for NAIs, but not significant differences were detected for NDIs; subsequently scaling heterogeneity of NAIs was higher for clayey than for medium textured soil. Differences in multifractal behaviour of NAIs and NDIs were consistent with a wider hysteresis loop of the medium texture soils compared to that of the clayey soils. Linear correlations were found between parameters $D_{-5}$ and ($D_{-5}$ - $D_5$) and clay content or SSA, which were positive and negative for NAIs and NDIs, respectively. Agronomical and environmental characterization of these soil groups with contrasting properties may be enhanced by evaluating SSA and by inspection of  NAIs and NDIs for multifractality.

**Key words:** nitrogen isotherms, soil specific surface, multifractals, texture, soil composition soil use.

**1. Introduction**

The quality of a soil, defined as its ability to perform a given function. In agroecosystems, the  suitability of a soil for a given  uses in agroecosystems, depends both on inherent or dynamic soil properties (Doran and Parker, 1994; Carter et al., 1997, Lal, 1998). Inherent soil properties such as particle size distribution, particle density, or soil mineralogy rely upon soil-forming factors, whereas dynamic soil properties, such as aggregate stability, water and nutrient status or bulk

density, are changing in response to soil use and management (Carter et al., 1997),  even if also may depend to some extent on  
[revised manuscript text omitted]

---

## Author Comment (AC3) · 17 Jul 2016

(1) Comments from Referee # 3 ( a) The multifractal analyses of nitrogen adsorption and desorption isotherms is obtained from soil samples of 6 different profiles in São Paulo State, Brazil and the effect of soil texture on the multifractal patterns of the isotherms is studied. A similar paper wrote by Paz-Ferreiro et al. (2013) presented a similar analysis with a different database from a different experimental site.

b) In my opinion the basis of the analysis that has been performed has serious problems, even if the potential value of the conclusions of the analysis in the context of soil sciences is high. This is because the analysis itself is not well founded from a numerical point of view. Specifically, it is not clear that any conclusion about the multifractal behaviour of a one dimensional series (temporal, spatial or of any other nature) with 41 or 52 data points could be reliable. In that respect I would put forward the work of A. Turiel, C. J. Pérez-Vicente and J. Grazzini entitle "Numerical methods for the estimation of multifractal singularity spectra on sampled data: A comparative study" (Journalof Computational Physics 216 (2006) 362–390). They study appropriate methods to assess multifractality over experimental discretized data and establish criteria to measure the confidence degree on the estimates. Manuscript npg 2105-72 do not complywith that criteria.

c) I understand that a mayor revision of the work should be undertaken in order to adapt the length of the series the authors use in their investigation to the type of analysis they propose. It is imperative to insure the consistency of the work done taking into account the potential benefit of the implications of this type of analysis for the "soil community".

(2) author's response Thank you very much for your valuable comments, The comments from Referee # 3 have been provided in three paragraphs, namely: a), b) and c). Author's responses have been organized following this paragraphs.

a) Multifractal analysis from Nitrogen adsorption (and some times also Nitrogen desorption isotherms) has been carried out before. As quoted in our manuscript (Page 3, Lines 23 to 26), in addition to Paz-Ferreiro et al. (2013) also Paz-Ferreiro et al. (2009, 2010), and Vidal-Vazquez and Paz-Ferreiro, (2012) analyzed multifractality of Nitrogren isohterms of soils, whereas Lado et al. (2013), addressed multifractality of artificial organoclays. Therefore, we agree with Referee #3 in that the analysis presented in not new. However, the data set and the aims in the previous work published by Paz- Ferreiro et al. (2013) and this work are not similar. In the former work the data set contained mainly clayey textured soil samples with a relatively wide range of organic matter content, which allowed assessment of the effect of organic matter. The main novelty of this work, is the contrasting texture of two groups of soil sample, i.e. medium versus clayey textured soils. So, the sampling strategy was chosen to evaluate

texture effects and to the best of our knowledge this is a new issue. Also we focused in agronomical and environmental characterization of the two soil types selected using multifractal analysis, which not addressed previously. This is explicitly stated at the end of the Introduction section and in the objectives (Page 3). We wrote: "Here, we hypothesized that analysis of the information contained in NAIs of NDIs of these two contrasting soil groups may provide further insight for its agronomical and environmental characterization.", and also (the objectives were): "to examine and to compare the scaling property of NAIs and NDIs in soils with contrasting texture".

b) Thank you very much for this comment. We agree that this is a substantial point. We performed multifarctal analysis using 5 partitions, as we would need 64 or more data points to use 6 partitions in the log-log relationship between partition function and scale. We are also aware of the study by Turiel et al. (2006) dealing with appropriate methods to assess multifractality over experimental discretized data and establish criteria to measure the confidence degree on the estimates. On the other hand, many authors stated that, on practice, one of the main drawbacks in multifractal analysis, when using the box counting method may be due to instabilities associated to the use of the Legendre transform. Please, note that we estimated the singularity spectrum using a direct method and we didn't employ the Legendre transform for this estimations. In this respect, also it should be pointed out that Turiel et al. (2006) discussed mainly the use of the Legendre transform and also they compared several methods including some methods, which perform trend removal prior to multifrcatal analysis (as for example wavelets) and therefore are much more demanding regarding the length of the data set. However, in practice many data sets used for multifractal analysis in several disciplines such as Geochemistry or Soil Sciences are much smaller. Please, see for example Wilson et al., .2016, Vadose Zone Journal, 15, doi:10.2136/vzj2015.04.0063, or Siquiera et al., 2014 Nonlinear Process. Geophys. 20, 529-541, in addition to the papers quoted in the previous paragraph. Indeed, the MF technique applied in this work is not an issue, the method is standard and the authors have published similar work involving other sites/properties. In our opinion, also it should be taken into account that we analyzed the scaling heterogeneity of several Nitrogen isotherm data sets and we focused in the use of the multifractal approach for assessing differences between soils with contrasting texture. It can be considered that to compare Nitrogen isotherms between different soil types is less demanding than to exhaustively characterize a unique soil data set (for example a single adsorption or desorption isotherm, using multifractals. In this respect the patterns of distribution of NAIs and NDIs assessed by multifractal analysis and new parameters obtained can be useful, as they are the result of physical processes occurring during adsorption or desorption of gas Nitrogen into soil aggregates. In addition, the q-moments choice is an issue which has been very much discussed in the literature (please, see our response to Referee # 2). This choice also depends from the number of data points used. The motivations are also related to the use of Legendre Transform (LT) and for the negative q-moments, in this last case the problems are well known. Admittedly, we decided to use q=+/-5 moments, but his aspect has not been stressed in our manuscript.

c) Also, thank you very much for stressing the need to assess clearly the viability of the analysis upon considering the number of data available. In fact we are aware that this is a critical point, and it has been also stressed by reviewers of previous works dealing with multifractal analysis of similar data sets. In the revised version we will try to clearly show why a multifractal analysis can provide improved knowledge that traditional analyses in the context and with the amount of data analyzed (again, please, see also our response to Referee # 2). In summary in the revised version (and also in previous work) we present arguments to prove unambiguously the strength of our results.

(3) author's changes in manuscript We tried to improve the basis of the analysis that has been performed and to increase the strength of the manuscript from a mathematical and numerical point of view. In this respect we paid more attention to the characterization of the multifractal spectra.

---

## Author Comment (AC4) · 12 Aug 2016

(1) Comments from Referee # 2 (29 February 2016)

Review of npg-2015-79 This manuscript presents multifractal analyses of nitrogen adsorption and desorption isotherms obtained on soil samples from 6 different soil profiles in São Paulo State, Brazil. There is very little that is new here, and the manuscript essentially repeats the study by Paz-Ferreiro et al. (2013), but in a different geographic area and with some less clayey soils. Although it lacks originality, I suppose there is some incremental new knowledge gained on the effects of soil texture on these types of multifractal analyses. Therefore, I recommend acceptance following minor revisions. My specific comments are itemized below (page #, line #):

[Figure]

1,12 & 4,27 – If SSA implies Euclidean geometry & method of moments analyses indicate multifractal geometry is it valid to present both sets of results for the same samples. Surely, the soil pore space is either Euclidean or multifractal, but not both. I am not sure that correlating SSA with generalized dimensions and Hölder exponents, as is done in Table 6, is a useful exercise.

1,18 – If the nitrogen adsorption and desorption isotherms are indeed multifractal, what are theimplications for soil pore size distributions? We have well-established physical models for Euclidean (e.g., random packing of uniform spheres) and monofractal (e.g., randomized Menger sponge) soils. Physically, what would a multifractal model for soil pore space geometry look like?

2,3 – Artificial is spelled incorrectly.

2,31 – Paz-Ferreiro and Vidal Vázquez (2012) does not appear in the references.

5,1 & Fig. 1 – It would be preferable to show the differential results rather than the cumulative plots in Fig. 1, which are very smooth and give absolutely no indication of multifractality.

6,5 – How many data points are needed to perform a robust multifractal analysis? Are 41 to 52 data points acceptable, and if so, according to what criteria?

6,6 – Report $R2$ values for the log-log linear regression analyses. Also, the residuals should be examined for the absence of trend.

6,7 – For a true multifractal the normalized measure versus scale is always linear on a log-log scale regardless of the subdivision level (k). Does the observation that these plots were non-linear for k < 1 imply that this is a pre-multifractal system operating over a limited range of scales, or is it due to the limited number of data points used in the analyses? 6,7 – Why was the range $-5 \leq q \leq +5$ chosen? Did the linearity of the plots change with q, and if so what are the implications of this? Again, for a true multifractal, linearity should not be a function of q.

(2) author's responses We appreciate your comments, as they raise an essential point related to the reliability of the multifractal method used to analyze the complex behavior of Nitrogen adsorption and desorption isotherms, and even the pertinence of our methodology. Referee # 2 contributed with a general comments and several specific comments itemized by: page #, line #. Author's responses have been organized following this scheme. General comment. A similar comment, about the novelty of our manuscript has been posted by reviewer # 3. We agree with reviewers #2 and #3 in that the analysis presented in not new, given that multifractal analysis from Nitrogen adsorption (and some times also Nitrogen desorption isotherms) has been carried out previously by several authors, as referenced in our manuscript. Note, however that the data set and the aims in the previous work published by Paz- Ferreiro et al. (2013) and the present work are different. In the former work the data set contained mainly clayey textured soil samples with a relatively wide range of organic matter content, which allowed assessment of the effect of organic matter. The main novelty of the present work, is the contrasting texture of two groups of soil sample, i.e. medium versus clayey textured soils. So, the sampling strategy was chosen to evaluate texture effects and to the best of our knowledge this is a new issue; this is explicitly stated at the end of the Introduction section and in the objectives. Also we focused in agronomical and environmental characterization of the two soil types selected using multifractal analysis, which not addressed previously.

Specific comments (by: page #, line #) - 1,12 & 4,27. This comment is very pertinent ad we totally agree with you in that the soil pore space is "either Euclidean or multifractal, but not both". Please, note that soil specific surface area (SSA) was obtained from the first part of nitrogen adsorption isotherms, i.e. for relative pressure (p/p0) values below 0.30, (as in this range a linearization has been proven to be possible); however multifractal analysis was performed for a wider range of relative pressures, i.e. $0 < p/p0 < 1$. Therefore data sets used for estimating SSA and to perform multifractal analysis are very different. This issue has been also addressed in Page 2, Lines 21 to 27. On the other hand, the exercise of correlation between SSA and multifractal parameters in

Table 6 (now Table 4 in the revised version of the manuscript), could be useful given the association between SSA and a number of soil physical and chemical properties.

- 2,3. The spelling mistake has been corrected.

- 2,31. Mistake has been corrected. Correct was Vidal Vázquez and Paz-Ferreiro, instead of Paz-Ferreiro and Vidal Vázquez.

- 5,1 & Fig. 1. Authors agree that cumulative plots of N2 isotherm "give absolutely no indication of multifractality". Indeed, differential curves are more irregularly distributed, and therefore may suggest multifractal behaviour. In fact, in previous work, differential plots NAIs as a function of either relative pressure (p/p0), (in Paz-Ferreiro et al., 2009) or as a function of pore sizes (Paz-Ferreiro et al., 2010) have been presented. However, in this particular manuscript we prefer to show and compare examples of the cumulative NAIs and NDIs for clayey and medium textured soils, because this better illustrate differences in the hysteresis loop between the two soil types. In our wok, the hysteresis loop has been found to be associated to the main differences in multifractality of the soils groups with contrasting textures studied.

- 6,5. Again, this is also a very important comment and also the question of the number of data points needed to perform a robust multifractal analysis also has been posed reviewer # 3. Next, we address this point and, please see also our answer to reviewer # 3. We are aware that that ideally and as recommended by Turcote 1992), for mutifractal analysis, the scale should be at least three orders of magnitude of the sampling intervals.(Turcotte, D.L. 1992. Fractals and Chaos in Geology and Geophysics. Cambridge University Press, Cambridge, 221 p). The larger the measurement scale and the number of data points available, the more reliable the results of mulifractal analysis. However in practice many data sets used for multifractal analysis in several disciplines such as Geochemistry or Soil Sciences are much smaller. Indeed, reliability of multifractal analysis depends on the number of points, but also on the range of q-moments used. Please, see additional comments in 6.7.

- 6,6. This is also a very valuable comment. A new Table showing R2 for the linear re-lationship of the partition function ïĄčïĂÍqïĂňïĄďïĂÍïĂăversus the scale, ïĄďïĂňïĂăhas been included (now, Table S3). This issue is now briefly addressed in the Methods (Page 6, Line 6) and in the Results and Discussion Sections (Page 7, Lines 22 to 27).

- 6.7. a) Does the observation that these plots were non-linear for k < 1 imply that this is a pre-multifractal system operating over a limited range of scales, or is it due to the limited number of data points used in the analyses?. In our opinion the answer to this question would need a review of the previous work on MFA of NAIs and NDIs published until now and also additional work. For example Paz Ferreiro et al. (2009) and Lado et al. (2013) reported non-linearity for k < 1, while Paz-Ferreiro et al.(2010) found linearity for 0 < k < 5. In the present manuscript we performed MFA for 32 horizons (64 adsorption and desorption isotherms) and for most of them we also found linearity for 0 < k < 5, but this was not the case at 3 of the studied data sets. however, in order to compare all the data sets studied we choose non-linearity for k < 1. This issue has been now briefly addressed in the Results and Discussion section. b) Why was the range -5 ≤ q ≤ +5 chosen?. In Literature this aspect is very discussed Again, the q-moments choice depends from the number of data points available. Again, using a smaller range of q moments results in reduction of large statistical errors, especially for q <0. Using a wider range of q moments better identifies differences between subsets. In previous papers dealing with multifractality of NAIs and NDIs, also different q moments ranges have been reported. We think that in our context q=+/-10 moments would be excessive, because of the small number of data points available.

(3) author's changes in manuscript - We edited the revised manuscript and corrected a large number of spelling and other language errors, in addition to the mistakes that reviewer # 2 showed in 2.3 and 2.31.

- We reported R2 values for the log-log linear regression analyses between partition function and scale, for several q moments, at the Supplementary Digital Content.

[Figure]

- Additional details about the way in which multifractal analysis has been performed (given the limited number of points in the available data sets) have been provided.

---

## Author Comment (AC5) · 12 Aug 2016

Dear Editor, Thank you very much for your comment on manuscript npg-2015-79. Also, we thank the reviewers for their comments on the manuscript. All the points raised by reviewers have been addressed. In particular, and as recommended, we addressed the following issues: - Novelty of the current manuscript and differences with a previous manuscript. - Methodological issues, mainly related to the pertinence of multifractal analysis. - Length of the paper, which has been reduced.

We hope that, at its present form, our manuscript would reach the high quality standards for publication on NPG

In behalf of the authors Dr. Jorge Paz-Ferreiro

Environmental Engineering Lecturer Department of Civil, Environmental and Chemical Engineering RMIT University, Melbourne, Australia